# Mitigation of Heat Propagation in a Battery Pack by Interstitial Graphite Nanoplatelet Layer: Coupled Electrochemical-Heat Transfer Model

Barbara Palmieri [1][ID], Fabrizia Cilento [1][ID], Ciro Siviello [2], Francesco Bertocchi [2], Michele Giordano [1] and Alfonso Martone [1],*

[1] Institute for Polymers, Composites and Biomaterials, Consiglio Nazionale delle Ricerche, P.le E. Fermi n.1, 80055 Portici, NA, Italy
[2] Jaber Innovation, 81030 Teverola, CE, Italy
* Correspondence: alfonso.martone@cnr.it

**Abstract:** The use of high thermal conductive materials for heat transfer is gaining attention as a suitable treatment for improving battery performance. Thermal runaway is a relevant issue for maintaining safety and for proficient employment of accumulators; therefore, new solutions for thermal management are mandatory. For this purpose, a hierarchical nanomaterial made of graphite nanoplatelet has been considered as an interface material. High-content graphite nanoplatelet films have very high thermal conductivity and might improve heat dissipation. This study investigates the effect of a thermally conductive material as a method for safety enhancement for a battery module. A numerical model based on the finite element method has been developed to predict the heat generation during a battery pack's charge and discharge cycle, using the Multiphysics software Comsol. The lumped battery interface generates appropriate heat sources coupled to the Heat Transfer Interface in 3D geometry. Simulation results show that the protection of neighbouring cells from the interleaved layer is fundamental for avoiding heat propagation and an uncontrollable heating rise of the entire battery pack. The use of graphite nanocomposite sheets could effectively help to uniform the temperature and delay the TR propagation.

**Keywords:** heat spreader; thermal runaway; graphite nanoplatelets; Li-Ion; finite element

## 1. Introduction

In recent years, high-energy, high-capacity lithium–ion batteries (LIB) have become widely used, including in electric vehicles (EVs), ships, and new alternatives for energy storage. During the LIBs' standard operating cycle, discharge at a high current will inevitably generate heat [1]. Heat accumulation in the battery pack could significantly reduce its useful life and give rise to the thermal runaway phenomenon (TR) [2], which may cause fire and even explosions [3]. Therefore, battery safety has gradually become a concern and a major technical issue limiting the applicability of LIBs [4].

The battery cell temperature must be regulated within a predefined operating range to sustain a rate of reaction considered healthy for the efficient operation of battery cells. The most significant causes of triggering TR are the unacceptable conditions derived from mechanical, electrical and thermal abuse. Mechanical abuse mainly occurs in automotive applications due to possible collisions that could damage the battery separator [5]. Electrical abuse is linked to the improper use of batteries, such as over-discharging, overcharging and external short circuits [6].

Thermal abuse, which is generally caused by an external ambient factor or local overheating, is also one of the principal causes of TR. When an LIB exceeds a specific temperature, its internal exothermic chain reactions are triggered [7]: (1) reaction between cathode and electrolyte; (2) thermal decomposition of electrolyte; (3) reaction between

electrolyte and anode; (4) thermal decomposition of the anode; (5) thermal decomposition of cathode [8–10]. This exothermic process generates combustible gases and results in the expulsion of the cell components [11]. Generally, the safety performance of LIBs should be improved by preventing the TR in three different ways: (1) enhance the electrolyte of LIBs to avoid burning; (2) improve the thermal stability of the electrode materials; (3) use new thermal management systems through active or passive heat dissipation mechanisms [12].

The risk of TR becomes even more severe in large battery packs, as the failure of a single cell could trigger TR propagation in the entire pack since the batteries are combined in series and parallel. When TR occurs for one cell in the battery pack, the amount of heat is rapidly transferred to the surrounding cells. At this point, the temperature of the neighbouring cell will be beyond its safe range, allowing the occurrence of the thermal runaway phenomenon in each cell.

Several studies are currently underway on the materials, structure and fabrication process of cathodes, anodes, and electrolytes to determine the triggering of TR mechanisms of batteries [13–16]. Current research on the uncontrolled thermal runaway propagation of lithium–ion battery packs is mainly carried out from the perspective of the external environment, prediction models, and cooling measurements. Liu et al. [17] studied the influence of a low-pressure environment on thermal runaway propagation in terms of external environment factors. It was found that the spread was slower than in the typical pressure environment, and the effective fire control time increased. On the other hand, Ren et al. [18] conducted experiments in both an adiabatic semi-enclosed environment and an open environment, concluding that the energy accumulated due to the difference in heat generation and heat dissipation conditions in different environments influences the rate of Thermal Runaway propagation. In terms of structural factors of the battery pack, studies have been carried out on how the increased space between cells affects the propagation of heat to adjacent cells. In addition, the impact of the battery's electrode connectors has also been investigated [19].

Several studies have been carried out to understand TR mechanisms due to thermal abuse by numerical simulations and experiments. Zhang et al. [20], through numerical simulation, have concluded that the principal heating generation in the TR process depends on the decomposition of electrolytes and the reaction between anode and electrolyte. Wang et al. [21] have focused their study on the flame characteristics, temperature and loss during the TR, also using a thin film heat flux sensor to study the heat dissipation of the battery surface to develop an effective battery thermal management system (BTMS). Kong et al. [22] have developed an electro-thermal model to analyse the effect of battery materials, external heating conditions and heat dissipation conditions on the TR process. The results highlighted how the bottom of LIB is the most sensitive to heat and should be better protected.

One way proposed by researchers to prevent the TR propagation is the use of insulation material or a cooling plate. Li et al. [23] assembled battery modules and aluminium plates used as heat spreaders to numerically study the effect of plate thickness on suppressing TR propagation. Moreover, keeping the LIB well ventilated and adding a thermally resistant layer could successfully prevent the TR propagation, as demonstrated by numerical simulation and experimental validation by Feng et al. [24]. As an alternative, the use of four different spacer materials, including air, an aluminium plate, a graphite composite sheet and aluminium extrusion, have been investigated by Yuan et al. [25], concluding that graphite composite and aluminium extrusions showed the best results.

Several studies have focused on reducing the propagation rate of heat by using thermal insulation materials, as heat spreaders, in order to increase the time interval of TR propagation. In fact, the critical stages that lead to the occurrence of the TR phenomenon can be identified as (1) the cell's self-extinguishing behaviour, which constitutes the temperature rise, due to an otherwise irrelevant discharge cycle; (2) a rapid drop in voltage causes a rapid temperature increase; (3) TR results uncontrollably, with understood heat generation [26]. Therefore, avoiding reaching the third stage is necessary to prevent un-

controlled cell explosion (Figure 1). The use of pyrolytic graphite sheets (PGS) with high thermal conductivity has successfully been employed for transporting heat out of the battery cells through conduction and dissipating this heat to the surrounding air through natural convection [27]. PGS is a synthetic material consisting of a uniform form of highly oriented graphite polymer sheet with a thermal conductivity of 600–800 Wm$^{-1}$K, and is stable up to 500 °C. Recently a new class of multifunctional films has been developed based on graphite nanoplatelets with high thermal conductivity and the ability to protect from high thermal fluxes [28].

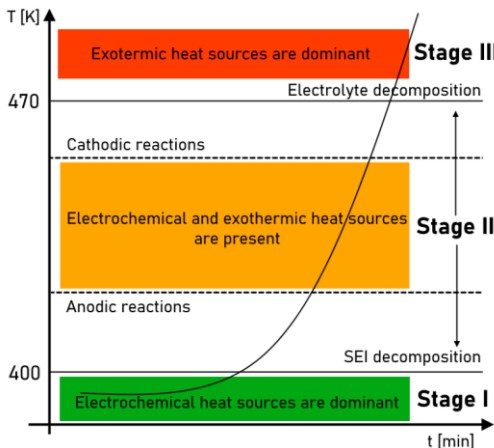

**Figure 1.** Critical stages that lead to the occurrence of the TR phenomenon.

In this study, a coupled electrochemical heat transfer has been developed to predict heat propagation in a battery pack. The model has been improved by including the heat generation kinetic related to the thermal runaway.

The model was set up by investigating the temperature distribution related to a single battery thermal abuse. The heat released from a triggered cell and its distribution inside the battery pack have been investigated, considering the effect of a GNP layer as a heat spreader. The results also indicate that for the safety design of the battery pack, the thermal path should be effectively controlled, and particularly the heating related to the exothermic reaction must be directed away from adjacent cells.

## 2. Development of the Thermal Model

A three-dimensional model has been built using COMSOL Multiphysics software to study the TR evolution of LIBs. LIBs with prismatic or pouch geometry are commonly used for electric and hybrid vehicles; the most commonly used lithium–ion batteries are the cylindrical 18,650 cells whose code name indicates the dimensions: 18 mm in width and 65 mm in length. In addition, the local temperature of the cylindrical-type LIBs is more inclined to overheat due to the worse contact condition with the refrigerant circulation tube than the prismatic LIBs [29], so the cylindrical 18,650 cell is preferably selected for simulation.

In this paper, the 18,650 LIB is selected as the research object, whose nominal capacity is 2.5 Ah, and the nominal voltage and cut-off voltage are 4.2 V and 1.37 V, respectively.

A typical internal structure of a LIB includes several thin layers, including an aluminium model current collector, the negative electrode, separator and positive electrode [13].

The single cell was considered in a simplified way as a homogeneous solid structure. It is assumed that the cell is locally heated in the open air. Convective heat transfer is considered for the effect of environmental cooling on the cell. The scheme of the lithium–ion battery pack is shown in Figure 1; in particular, the system consists of six cylindrical cells placed inside a battery box (Figure 2).

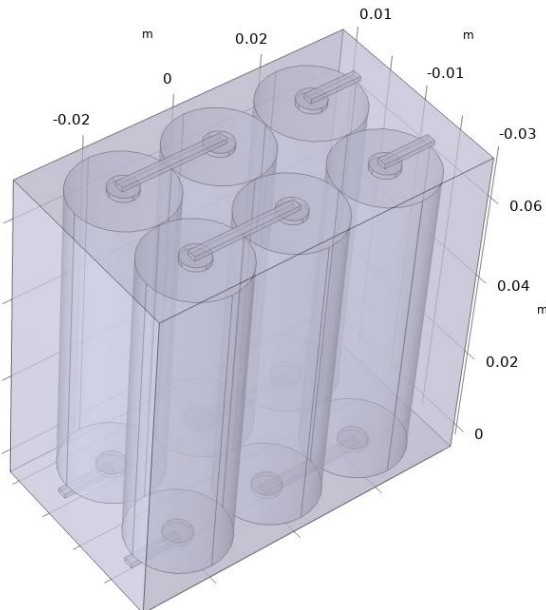

**Figure 2.** Scheme of the battery pack.

The geometric parameters of the simulated battery and the PVC holder of the battery pack are given in Table 1, while their physical parameters are listed in Table 2, which are based on the experimental data reported in the literature.

**Table 1.** Geometric parameters were used in the simulation.

| Nomenclature | Parameters | Value |
|---|---|---|
| Battery diameter | L (mm) | 21 |
| Battery height | H (mm) | 70 |
| Battery distance | d (mm) | 0.1 |
| Terminal thickness | t_term (mm) | 1 |
| Terminal radius | r_term (mm) | 3 |
| Serial connector width | w_sc (mm) | 2 |
| Serial connector height | h_sc (mm) | 1 |
| Parallel connector height | h_pc (mm) | 0.5 |
| Parallel connector width | w_pc (mm) | 1 |
| PVC holder height | h_hold (mm) | 75 |
| PVC holder length | l_hold (mm) | 150 |
| PVC holder width | w_hold (mm) | 20 |

**Table 2.** Thermo-physical parameters of materials used in the simulation.

| Nomenclature | Parameters | Value | Reference |
|---|---|---|---|
| Density of battery | $\rho_{batt}$ | 2055 (kg/m$^3$) | [30] |
| Specific heat capacity of the battery | $C_{p, batt}$ | 1400 (J/kg·K) | |
| Radial thermal conductivity of the battery | $k_{T,batt,r}$ | 0.897 (W/m·K) | [30] |
| Axial thermal conductivity of the battery | $k_{T,batt,ang}$ | 29.56 (W/m·K) | [30] |
| The density of battery holder | $\rho_{hold}$ | 1100 (kg/m$^3$) | |
| Thermal conductivity of battery holder | $k_{T,hold}$ | 0.03 (W/m·K) | |
| Specific heat capacity of the battery holder | $C_{p, hold}$ | 1400 (J/kg·K) | |
| Density of connector | $\rho_c$ | 2700 (kg/m$^3$) | |
| Thermal conductivity of battery connector | $k_{T,box}$ | 204 (W/m·K) | |
| Specific heat capacity of the battery connector | $C_{p, box}$ | 880(J/kg·K) | |

Three different models have been developed to describe the battery pack's overall behaviour, including the passive treatment based on graphite nanoplatelets: electrochemical behaviour, heat transfer of the battery pack, and TR kinetic implementation.

The following paragraphs will introduce the geometric structure, model parameters, boundary and initial conditions, and solution steps.

### 2.1. Electrochemical Model

Two different physical models have been coupled in this model: lumped battery and heat transfer. The electrochemical phenomena inside the LIB due to the charge and discharge phases could be simulated through the lumped battery module.

$$E_{cell} = E_{OCV}(SOC, T) + \eta_{IR} + \eta_{act} \tag{1}$$

$$I_{1C} = \frac{Q_{cell,0}}{3600\,[s]} \tag{2}$$

$$I_{cell} = I_{app} \tag{3}$$

$$V_{cell} = \int_{\Omega_{cell}} d_{vol}\partial\Omega \tag{4}$$

$\eta_{IR}$ is the value of ohmic overpotential that depends on the ohmic overpotential at 1C, where $C$ is the battery discharge rate. The $I_{1C}$ is a function of the battery capacity.

The $I_{cell}$ is the value of current applied to the battery and is an input value; in this paper, a charge-discharge cycle of the battery has been applied, so the current has the trend shown in Figure 3; the open-circuit case was also considered, and in this case, the input current value is 0 A.

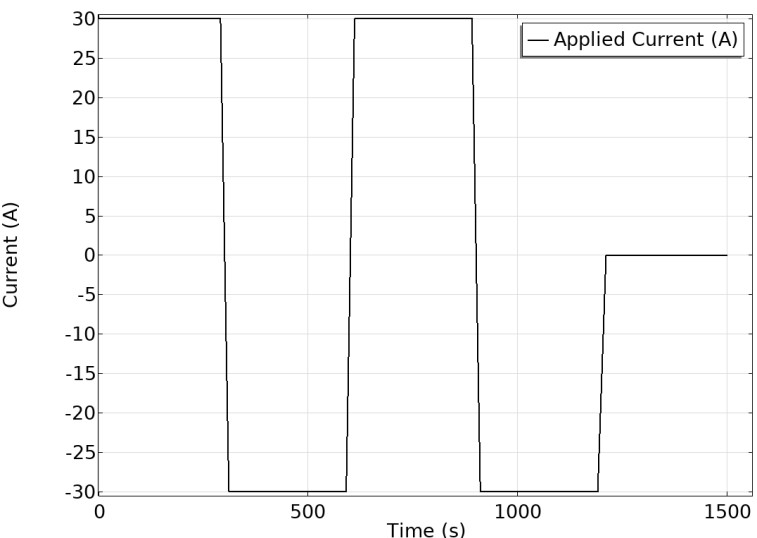

**Figure 3.** Charge-discharge cycle applied at the battery.

$E_{OCV}$ is the battery's open circuit voltage (*OCV*)

$$E_{OCV}(SOC, T) = E_{OCV,ref}(SOC) + \left(T - T_{ref}\right)\frac{\partial E_{OCV}(SOC)}{\partial T} \tag{5}$$

$E_{OCV,ref}$ is the open circuit voltage at the reference temperature depending on the battery's state of charge (*SOC*); it is an intrinsic property of the battery [8], and its trend is reported in Figure 4.

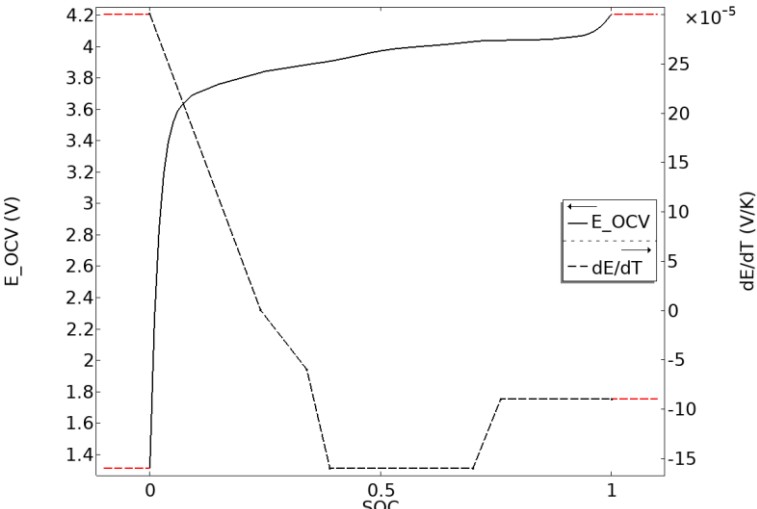

**Figure 4.** Curves of $E_{OCV}$ vs. *SOC* at the reference temperature and of the *SOC* derivative with respect to the reference temperature vs. the *SOC*.

When the cell is exposed to the electric load $I_{cell}$, heat will be generated inside the cell due to the reversible and irreversible processes in the cathode, anode, and the electrolyte of the LIB.

$$Q_{el-chem} = Q_{rev} + Q_{irrev} \tag{6}$$

$$Q_{rev} = I_{cell} \cdot T \cdot \frac{\partial E_{OCV}}{\partial T} \tag{7}$$

$$Q_{irrev} = I_{cell} \left( E_{OCV,ref} - E_{OCV} \right) \tag{8}$$

Reversible heat results from the change in entropy associated with the chemical reactions occurring within the battery. It is, therefore, also called reaction heat or entropic heat. On the other hand, irreversible heat comes from heat due to the polarisation of the active phase and heat due to the Joule effect. The first is the heat generated due to the deviation between the open circuit and operating voltage. The second is the energy dissipated due to the resistance of the solid phase and electrolyte to crossing lithium ions.

Once the electrochemical model's boundary and initial conditions are imposed, the thermal element has been considered. The energy balance in the battery cell and the box can be expressed as follow [30]:

$$\rho c_p \left( \frac{\partial T}{\partial t} \right) = \lambda \frac{\partial^2 T}{\partial x^2} + Q_{el-chem} + Q_{TR} - Q_{conv} \tag{9}$$

where $Q_{TR}$ denotes the heat generated by the side reaction due to the TR, and $Q_{conv}$ signifies the heat lost due to the convective heat flux from the environment.

In this model, the energy balance is modified as follows:

$$\rho c_p \left( \frac{\partial T}{\partial t} \right) = \frac{1}{r} \frac{\partial}{\partial r} \left( \lambda_r r \frac{\partial T}{\partial r} \right) + \frac{1}{r^2} \frac{\partial}{\partial \varphi} \left( \lambda_\varphi \frac{\partial T}{\partial \varphi} \right) + \frac{\partial}{\partial z} \left( \lambda_z \frac{\partial T}{\partial r} \right) + Q_{el-chem} + Q_{TR} - Q_{conv} \tag{10}$$

The battery's thermal conductivity is anisotropic, so the radial and the axial conductivity are defined as follows:

$$k_r = \frac{\sum_i L_i}{\sum_i \frac{L_i}{k_{T,i}}} \tag{11}$$

$$k_a = \frac{\sum_i L_i k_{T,i}}{\sum_i L_i} \tag{12}$$

A solid domain taking into account the anisotropic properties was made for each cell, and the corresponding cylindrical coordinate system was then selected.

An additional set of domains was made for the connectors and the holder in which the batteries are placed. In this case, the global coordinate system was considered. In addition, the convective heat flow was considered, simulating still airflow (convection heat transfer coefficient of 5 W/m²·K [31,32]) on the external surfaces of the battery holder (Figure 5).

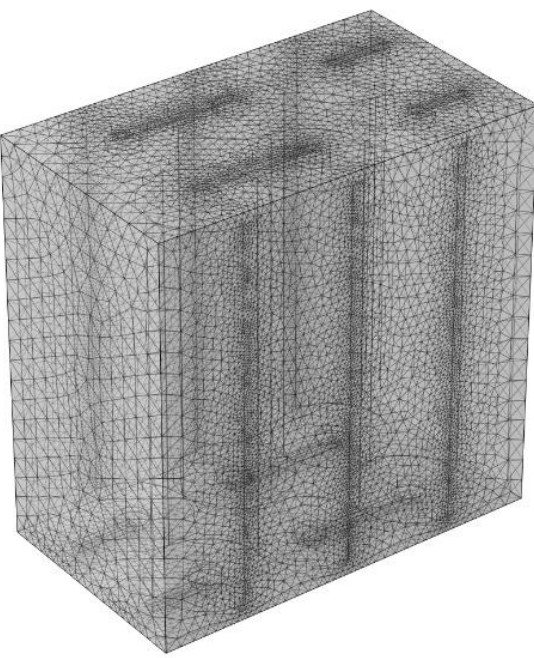

**Figure 5.** Model mesh realised.

Having set the boundary conditions, the mesh of the entire system was then constructed.

Given the cylindrical geometry of the individual cells, an extruded mesh with a fixed number of elements was built for these domains. For the remaining domains, the mesh was composed of unstructured tetrahedral elements.

### 2.2. Thermal Management by Graphene Rich Layers

In this numerical model, only the thermal part was considered to study the effective advantage of graphene-rich layers as spacer material when a single cell of the entire battery pack was subjected to uncontrolled temperature rise.

Therefore, an initial temperature equal to 130 °C was applied only to the cell to avoid irreversible TR (Figure 6).

The temperature distribution within the entire battery pack was evaluated to compare the result with a case in which a GNP sheet is used as spacer material.

The geometric and thermal properties of the graphene layer are reported in Table 3 and are based on experimental data reported in the literature [28].

**Table 3.** Thermo-physical parameters of GNP layer used in the simulation.

| Nomenclature | Parameters | Value | Reference |
|---|---|---|---|
| Density of GNP | $\rho_{GNP}$ | 2055 (kg/m³) | [28] |
| Specific heat capacity of GNP | $C_{GNP}$ | 1400 (J/kg·K) | [28] |
| In-plane thermal conductivity of GNP | $k_{//,\,GNP}$ | 0.897 (W/m·K) | [28] |
| Cross plane thermal conductivity of GNP | $k_{\perp,\,GNP}$ | 29.56 (W/m·K) | [28] |
| GNP thickness | $\tau_{GNP}$ | 100 (μm) | |

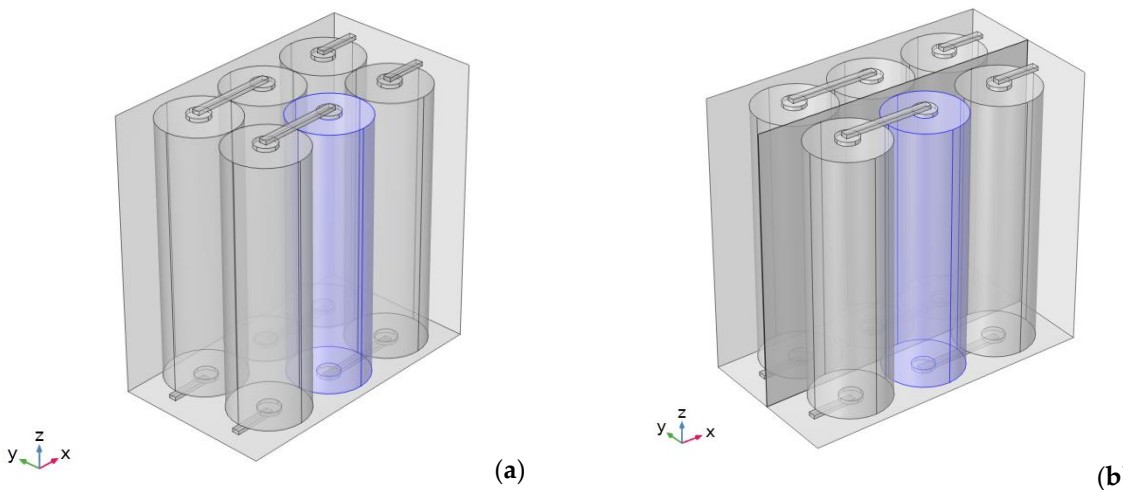

**Figure 6.** Triggered battery with a $T_{in}$ of 130 °C: (**a**) without GNP layer; (**b**) with GNP layer.

### 2.3. Heat Generation Due to Thermal Runaway

The subsequent model development was meant to consider, apart from heat generation due to the discharge-charge cycle, the additional source of heat related to exothermic reactions, and thus the phenomenon of thermal runaway.

In particular, the initial stages, i.e., decomposition reactions, were considered. The decomposition reactions begin in the temperature range of 80–120 °C [33].

To simulate thermal runaway, the heat equation has been coupled with ordinary differential equations (ODEs), which describe the time evolution of the concentration of exothermic reactions based on an Arrhenius-type law. Thus, the exothermic heat source is given by the equation:

$$Q_{TR} = H_i \cdot \frac{da}{dt} \tag{13}$$

$$\frac{da}{dt} = A_i \, \exp\left(-\frac{E_{a,i}}{RT}\right) \cdot \alpha(T, t) \tag{14}$$

where $H_i$ represents the enthalpy of reaction in J/g, $A_i$ is the rate factor in 1/s, $E_{ai}$ is the activation energy in J/mol, and $R$ is the universal gas constant.

The values imposed for these parameters were obtained from the literature and are reported in Table 4.

**Table 4.** Kinetic parameters used for the modelling of the TR.

| Nomenclature | Parameters | Value | Reference |
|---|---|---|---|
| Pre-exponential factor | $A_i$ | $1.67 \times 10^{15}$ (1/s) | [34] |
| Reaction Enthalpy | $H_i$ | $2.57 \times 10^5$ (J/kg) | [34] |
| Activation Energy | $E_{ai}$ | $1.35 \times 10^5$ (J/mol) | [34] |
| Universal Gas Constant | R | 8.314 (J/mol·K) | [34] |

A discharge–charge cycle, with a total duration of 1500 s, was also considered for that model, adding $Q_{TR}$ (13) as an additional heat source applied to all batteries' cells.

## 3. Results

### 3.1. Effect of the Charge-Discharge Cycle on the Battery Pack

The temperature increased during regular battery operation up to 70 °C. Although no external factors can lead to uncontrolled temperature increases, the discharge–charge cycle can lead to a rise in the temperature of individual cells. In addition, during the open-circuit phase, i.e., zero applied current, the temperature decreases due to the cell's self-extinguishing behaviour. Figure 7a shows the current and relative voltage trends during

the discharge–charge cycle of the batteries. Figure 7b, however, shows the temperature trend within the three pairs of cells as the current changes.

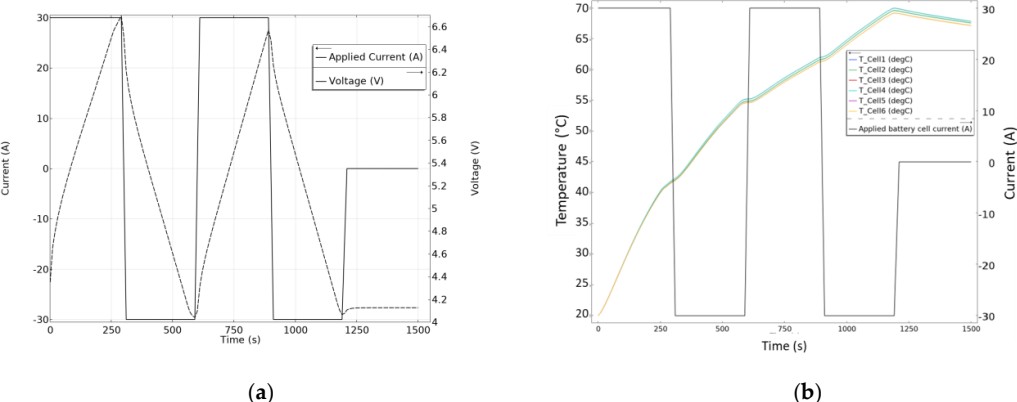

(**a**)  (**b**)

**Figure 7.** (**a**) Change in current and voltage during the discharge–charge cycle of batteries; (**b**) temperature rise vs. applied current.

By analysing the 3D temperature field shown in Figure 8, it is possible to see the temperature trend within the entire module at different instants of time (0, 500, 100 and 1500 s). At the initial moment, the whole module is at an ambient temperature of 20 °C. In contrast, after 500 s, and thus at the end of the first discharge cycle, the cell temperature increases to about 50 °C. In contrast, the temperature inside the PVC holder remains around 20 °C. The presence of the aluminium connectors also constitutes a thermal bridge between the cells connected in series and thus contributes to heat propagation.

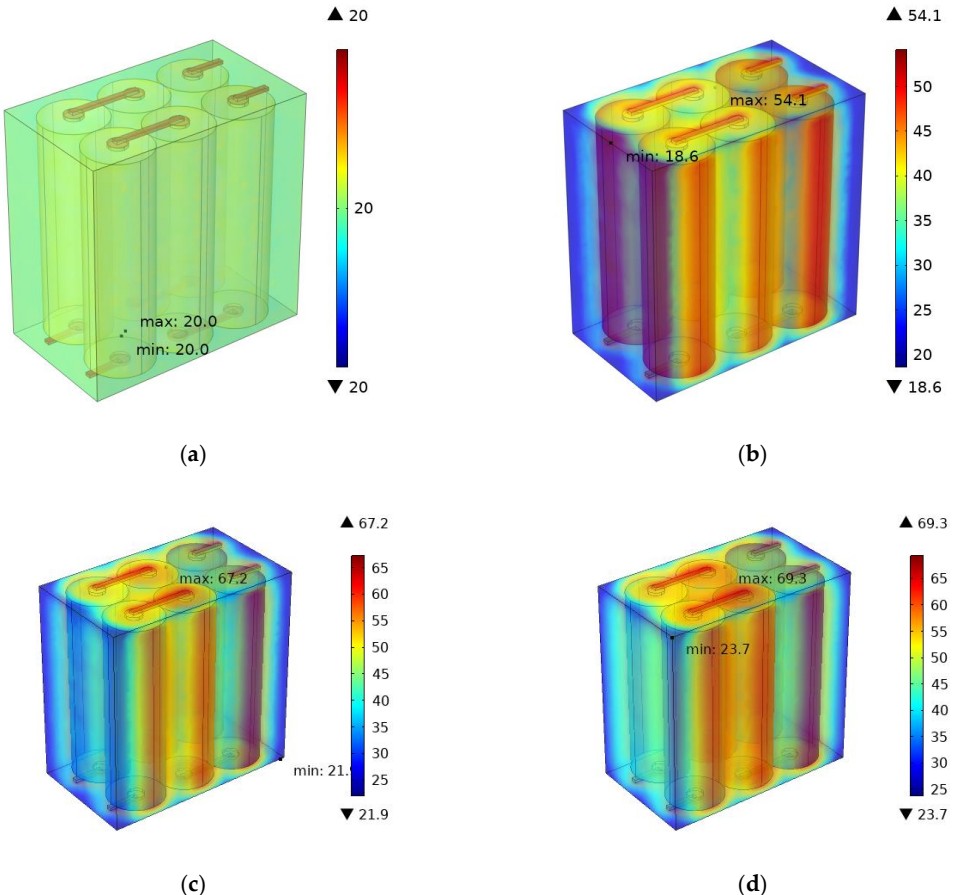

(**c**)  (**d**)

**Figure 8.** Temperature distribution inside the battery pack at different instants of time: (**a**) t = 0 s; (**b**) t = 500 s; (**c**) t = 1000 s and (**d**) t = 1500 s.

After 1000 s, i.e., at the end of the charge cycle, the pack's temperature increases, although at a slower rate than in the previous discharge cycle. Finally, after 1500 s, the maximum temperature inside the battery pack was about 70 °C.

### 3.2. Implementation of the Thermal Runaway Phenomenon

Figure 9 shows the reaction rate made explicit by (10) and the amount of heat generated by the decomposition reaction (Equation (13)).

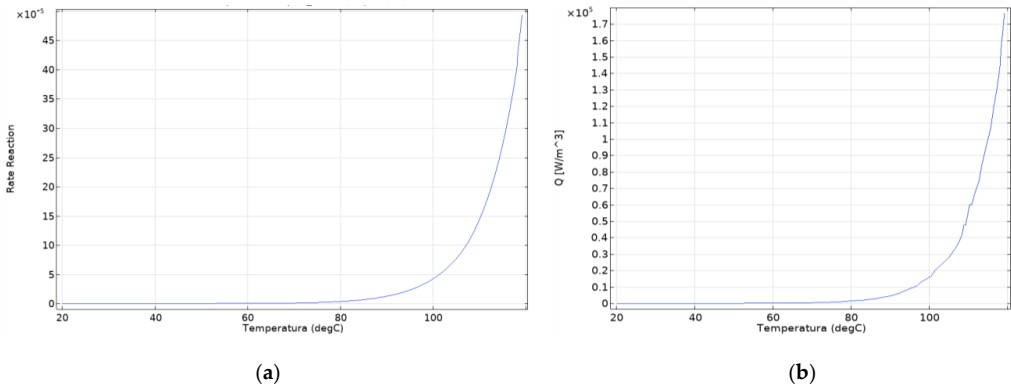

(**a**)    (**b**)

**Figure 9.** (**a**) Curve of reaction rate vs. temperature (Equation (9)); (**b**) heat generation by decomposition reactions (Equation (10)).

Figure 10 shows the temperature distribution inside the battery pack for four different time instants (0, 500, 1000 and 1500 s).

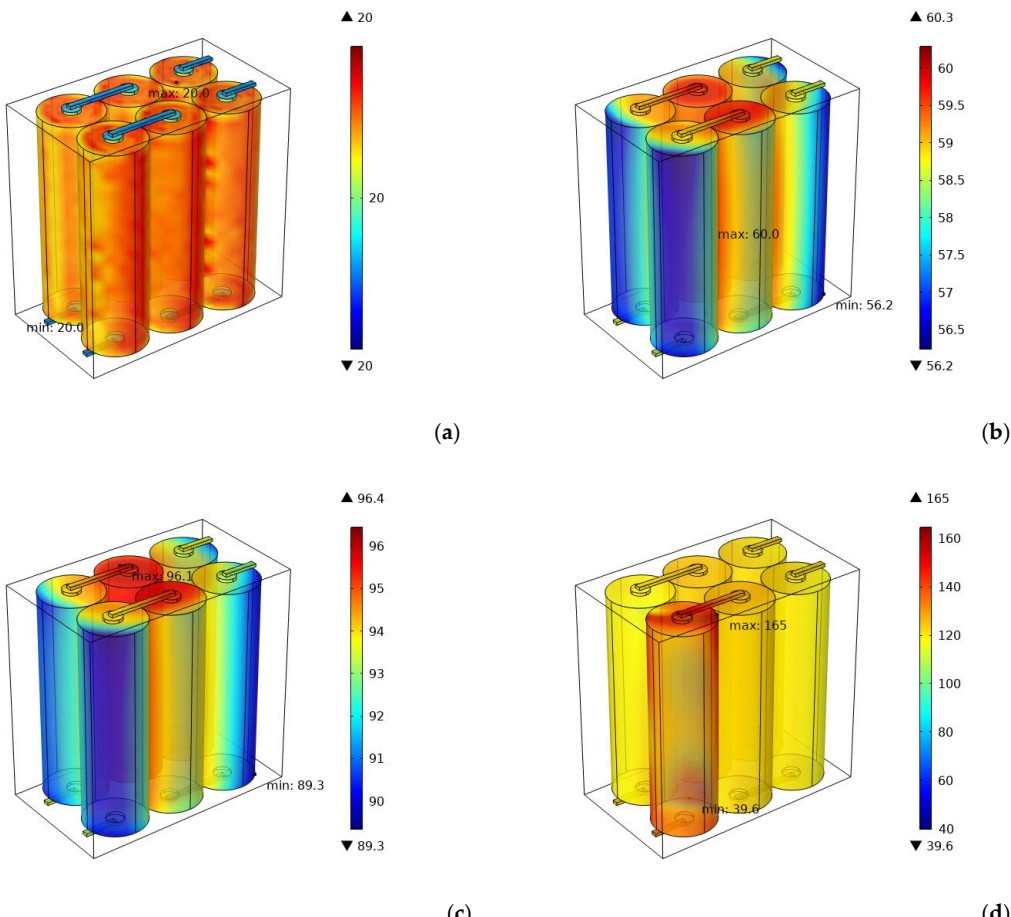

(**a**)    (**b**)

(**c**)    (**d**)

**Figure 10.** Temperature distribution inside the battery pack as a result of the decomposition reactions: (**a**) instant t = 0; (**b**) t = 500 s; (**c**) t = 1000 s and (**d**) t = 1500 s.

In line with what has been reported in [34], above 80 °C, the decomposition reaction accelerates to a sudden and uncontrolled battery temperature value and the occurrence of the thermal runaway phenomenon. The temperature of the batteries increases by about 65 °C in less than 200 s (Figure 11).

The model developed was thus able to replicate the thermal profile trend studied by Mishra et al. [34], in STAGE I & II, before the occurrence of phase III, where the decomposition reactions of the battery elements become uncontrollable.

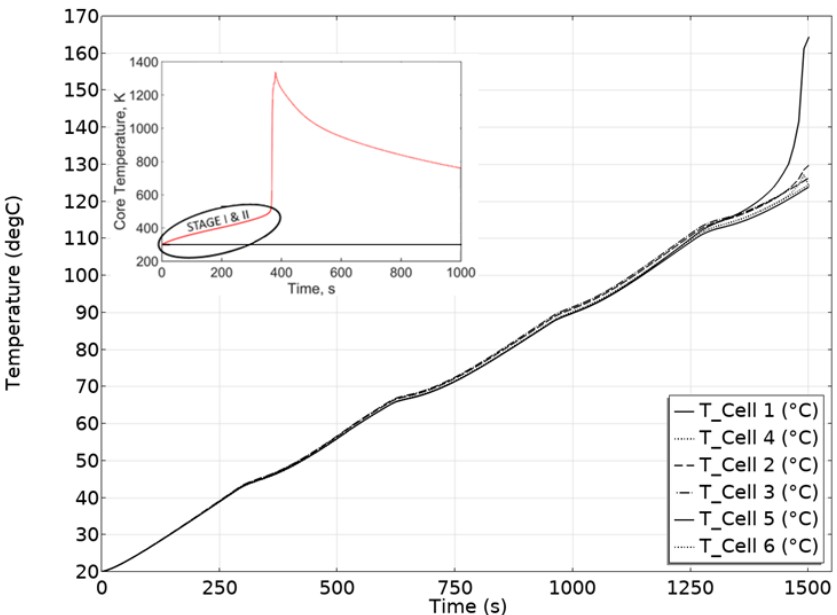

**Figure 11.** Temperature vs. time prediction; the model includes the TR kinetic. The inset picture reproduces the experimental temperature trend when TR is incipient, adapted from [34].

### 3.3. Thermal Management by GNP Nanocomposites

In Figure 12, the effects of the thermal distribution and the heat spreader of the GNP layer are depicted.

The presence of the GNP layer promotes the warm-up of the cell placed orthogonal to the triggered direction, while those directly adjacent are not affected by heating.

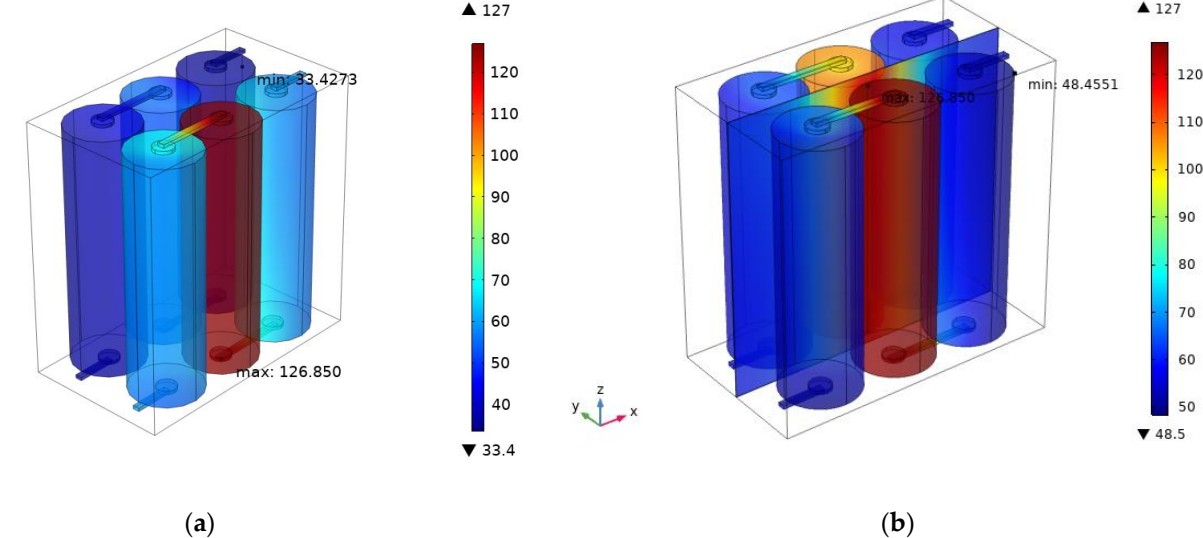

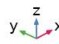

(**a**)            (**b**)

**Figure 12.** Temperature distribution inside the battery pack after 1500 s: (**a**) no thermal conductive layer; (**b**) by considering the GNP layer as heat spreader with the ambient.

This different distribution of heating in-plane and cross-plane, as evident from Figure 13, of the GNP layer, depends on the anisotropy of this material [35].

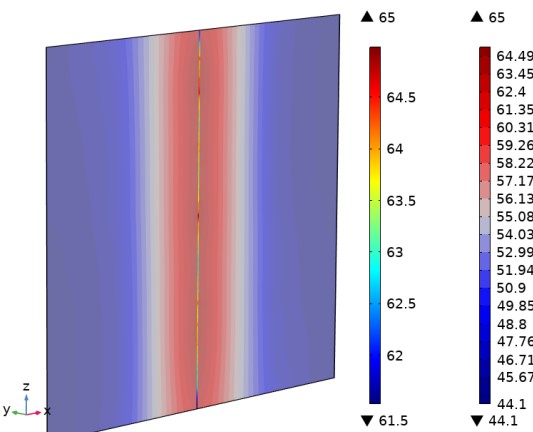

**Figure 13.** Thermal distribution in plane and cross-plane of the GNP layer.

## 4. Discussion

### 4.1. Dissipating Heat by GNP Film

To evaluate the heat spreading effect of the GNP nanolayer, a new electrochemical simulation has been developed considering the presence of the GNP layer as interleaved material between the batteries.

Figure 14a reports the temperature rise in the batteries due to the charge–discharge cycle, while in Figure 14b is depicted the thermal distribution inside the GNP layer.

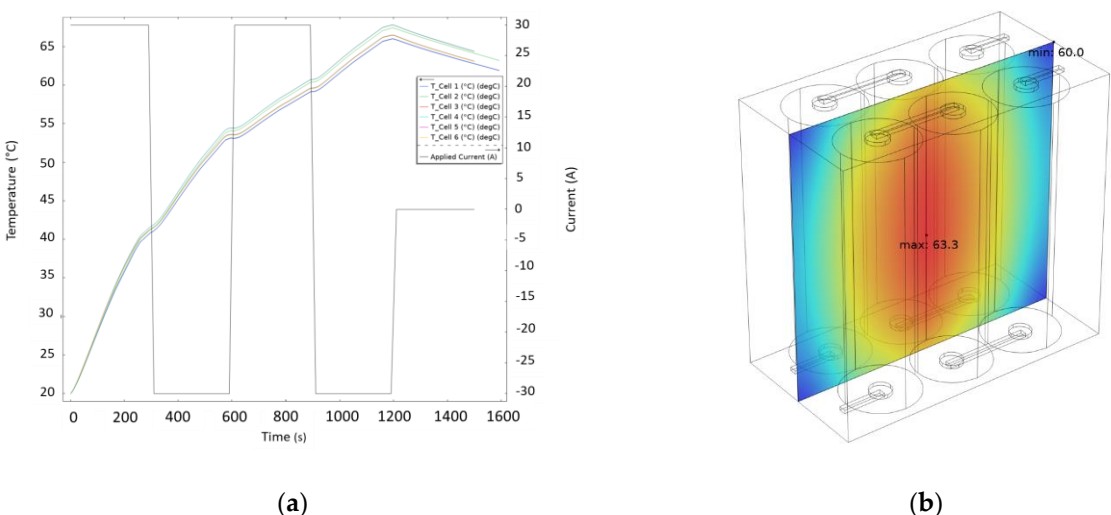

(**a**)                                                                                           (**b**)

**Figure 14.** (**a**) Temperature rises inside the batteries vs. applied current for the model with the GNP as heat spreader; (**b**) temperature distribution inside the GNP layer.

The effectiveness of the GNP layers as heat spreaders is evident in Figure 15, where the thermal distribution of the battery pack is reported without (Figure 15a) and with (Figure 15b) the nanocomposite layer. In particular, the presence of the heat spreader does not affect the maximum temperature reached, but helps to achieve a more uniform temperature distribution, improving the cooling.

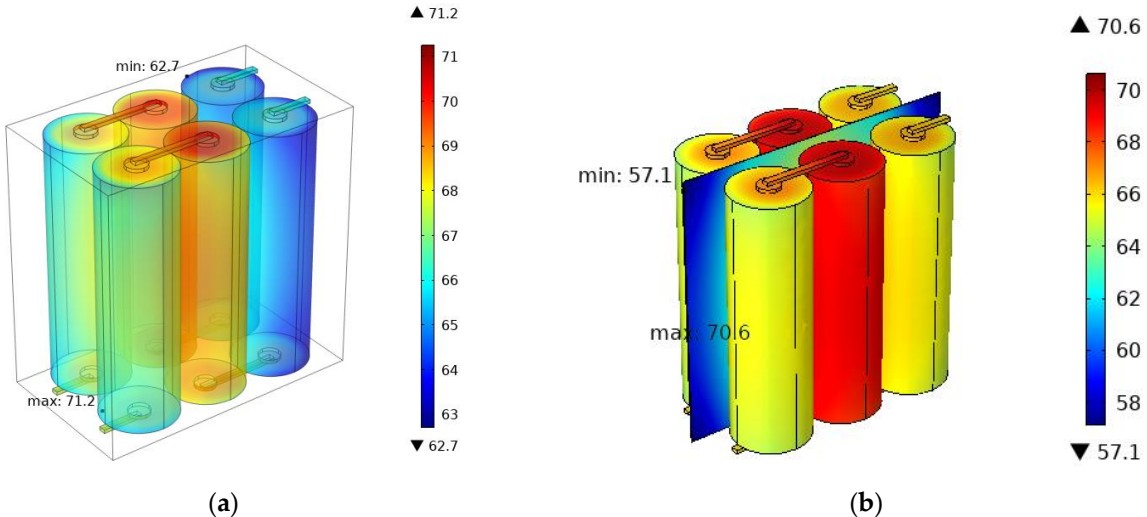

(**a**)  (**b**)

**Figure 15.** (**a**) Temperature distribution inside the battery pack without the heat spreader; (**b**) temperature distribution inside the battery pack with the heat spreader.

*4.2. Thermal Runaway Protection by Nano Laminate*

This paper also investigates the effect of the presence of the GNP nanolayer on the TR processes. Figure 16 shows the temperature distribution inside the batteries due to the TR phenomenon without and whit the nanocomposite layer. By comparing 16a,b, it is evident that the maximum temperature reached by the batteries is lower due to the presence of the heat spreader (30 °C). It is worth noting that without the GNP layer, all the batteries reach a critical temperature. In contrast, the thermally conductive layer acts as temperature mitigation by removing heat from the environment.

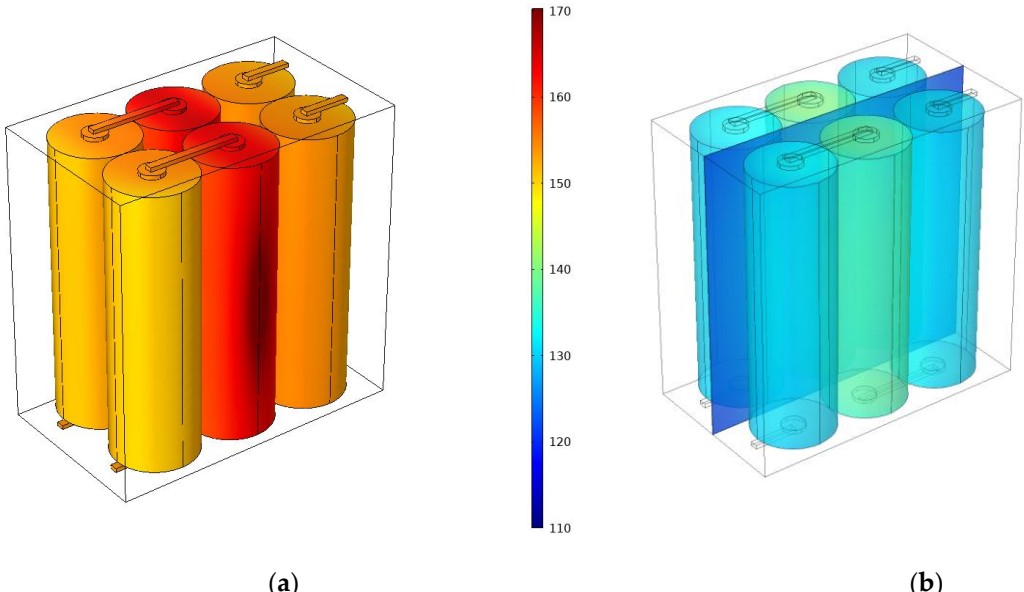

(**a**)  (**b**)

**Figure 16.** (**a**) Temperature distribution inside the battery pack without the heat spreader; (**b**) temperature distribution inside the battery pack with the heat spreader.

Figure 17 compares the temperature profile of a point in the middle section without any treatment and in the case of a GNP film able to transfer heat on external ambient on a strip 10 mm wide. The introduction of a thermal conductive layer aims to distribute heat in the volume, mitigating the maximum temperature reached (110 °C). The temperature field reports a reduction of 20 °C (time 1450 s) reducing the risk of triggering TR onset.

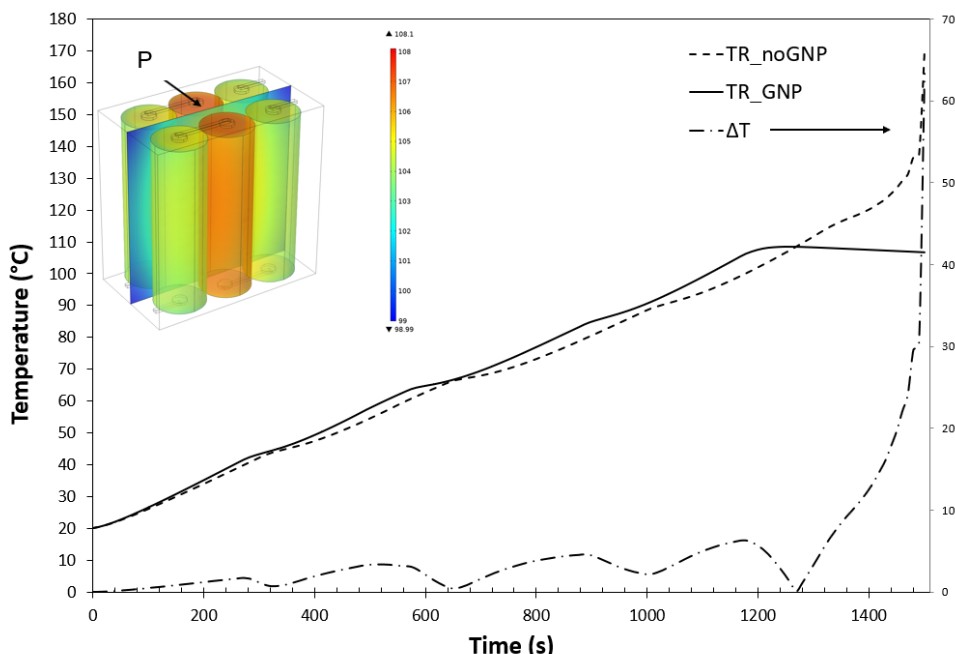

**Figure 17.** Maximum temperature curve vs. time, for the case with and without the heat spreader.

## 5. Conclusions

The initiation and propagation of thermal instability involve multiple nonlinear thermal processes. Therefore, it is necessary to use a numerical model that accounts for these processes and their interaction to predict whether TR propagation will occur. In the present work, a coupled electrochemical heat transfer model was set up to investigate the effectiveness of using a thermal conductive layer for mitigating the temperature field. To account for the thermal response of a battery pack, the model includes:

- The effect of charge–discharge on the temperature distribution.
- Thermal runaway modelling by including the heating kinetic.

Using graphite nanoplatelet-rich films as an interleaved layer resulted in an effective temperature mitigation strategy, since it acts as a temperature mitigation layer when connected to the ambient.

**Author Contributions:** B.P., F.C. and A.M.: Conceptualization, Methodology; B.P. and C.S.: data curation; B.P. and A.M.: software; B.P., F.C. and C.S.: Validation; F.B. and M.G.: supervision; F.B.: funding acquisition; B.P. and F.C. Writing—original draft; F.B., M.G. and A.M.: Writing—review & editing. All authors have read and agreed to the published version of the manuscript.

**Funding:** This research was carried out in the framework of the project MUSAICO, grant number F/190014/01-02/X44, founded by the Italian Government.

**Data Availability Statement:** Not applicable.

**Conflicts of Interest:** The authors declare no conflict of interest.

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
