# Peer review of "Mitigation of Heat Propagation in a Battery Pack by Interstitial Graphite Nanoplatelet Layer: Coupled Electrochemical-Heat Transfer Model"

_jcs, doi:10.3390/jcs6100296_

Round 1

Reviewer 1 Report

Review of the paper “Mitigation of heat propagation in a battery pack by interstitial 2 graphite nanoplatelet layer: Coupled electrochemical-heat 3 transfer model” by B. Palmieri et al.

The authors consider the possibility to use graphite spacers in a battery pack in order to improve heat dissipation. In order to verify the possibility they develop a 3D FEM model and examine the effect of spacers on overall  performance of the device in detail. The subject of study might be appropriate for Journal of Composites Science.

There are several problems about this manuscript, most often related to poor language, but these are not the only ones.

 Line 124 “the cylindrical 18,650 cell” is a bit unclear, did the authors mean “ A system consisting of 18,650 cylindrical cells”? this should be clarified at the first occurrence, later the authors may use the abbreviation “the 18,650 LIB”

“interstitial”  in different parts of the manuscript – consider the use of “spacer”, the reviewer understands the word, but thinks it is more used in medicine

„spreading heat” (abstract line 11) is rather unusual, maybe “heat transfer” is more common

“proficient the emloyment” (line 13) – “for” missing

Line 16 maybe “might improve heat dissipation”

Line 87 “Moreover, also” – the first word “moreover” already implies that an additional issue is taken into account, thus “also” is a synonym here, please remove.

Line 89 should be “As an alternative” “an” missing

Line 147 “three … models”

Line 151 “the geometric structure and the model parameters”

Line 154 “two different physical models” (you cannot couple “physics” which is a scientific discipline)

Line 195 “is modified”

Line 198 “as follows”

Line 204 “the convective heat flow”

Line 218 the correct syntax should be “Therefore, the initial temperature equal to 130 Degs was applied only to the cell, in order to avoid ….”

Line 230 awkward syntax, maybe “The subsequent model development was meant to consider, apart from heat generation …, the additional source ….”

Line 285 maybe better “in Reference [34]”

Line 297 “ In Figure 12 the effects of thermal distribution and the heat spreader ? (maybe dissipation) of the GNP layer are depicted”

Line 323 “the cooling”, “cool down” is a phrasal verb

No comment on what has been depicted in Figure 17. Do the authors consider the relatively small differences in max. temp. vs. time profiles so important and relevant at some time instants so important, that the whole effort aimed at improvement of heat dissipation of the battery pack is justified? How about the costs of assemblage of relatively thin and most probably expensive material within the modules?

Author Response

The authors consider the possibility to use graphite spacers in a battery pack in order to improve heat dissipation. In order to verify the possibility, they develop a 3D FEM model and examine the effect of spacers on overall performance of the device in detail. The subject of study might be appropriate for Journal of Composites Science

  1. There are several problems about this manuscript, most often related to poor language, but these are not the only ones

The authors have assessed the errors reported in the manuscript and have performed extensive editing of the English language.

  1. Line 124 “the cylindrical 18,650 cells” is a bit unclear, did the authors mean “ A system consisting of 18,650 cylindrical cells”? this should be clarified at the first occurrence, later the authors may use the abbreviation “the 18,650 LIB”

The code 18650 is a standard nomenclature for rechargeable batteries, the manuscript was modified as follows (line 123 and subsequent) when first we used the 18650 codename:

“…. the most used lithium-ion batteries are the cylindrical 18650 cells whose code name indicates the dimensions: 18 mm in width and 65 mm in length.”

  1.  “interstitial” in different parts of the manuscript – consider the use of “spacer”, the reviewer understands the word, but thinks it is more used in medicine

The term “interstitial” has been replaced according to the reviewer suggestion.

  1. „spreading heat” (abstract line 11) is rather unusual, maybe “heat transfer” is more common

The statement has been corrected according to the reviewer suggestion

  1. “proficient the emloyment” (line 13) – “for” missing.

Fixed

  1. Line 16 maybe “might improve heat dissipation”

Fixed

  1. Line 87 “Moreover, also” – the first word “moreover” already implies that an additional issue is taken into account, thus “also” is a synonym here, please remove

Fixed

  1. Line 89 should be “As an alternative” “an” missing

Fixed

  1. Line 147 “three … models”
  1. Fixed

  1. Line 151 “the geometric structure and the model parameters”

Fixed

  1. Line 154 “two different physical models” (you cannot couple “physics” which is a scientific discipline)

Fixed

  1. Line 195 “is modified”

Fixed

  1. Line 198 “as follows”

Fixed

  1. Line 204 “the convective heat flow”

Fixed

  1. Line 218 the correct syntax should be “Therefore, the initial temperature equal to 130 Degs was applied only to the cell, in order to avoid ….”

Fixed

  1. Line 230 awkward syntax, maybe “The subsequent mode development was meant to consider, apart from heat generation …, the additional source ….”

Fixed

  1. Line 285 maybe better “in Reference [34]”

Fixed

  1. Line 297 “ In Figure 12 the effects of thermal distribution and the heat spreader ? (maybe dissipation) of the GNP layer are depicted”

Fixed

  1. Line 323 “the cooling”, “cool down” is a phrasal verb

Fixed

  1. No comment on what has been depicted in Figure 17.

The manuscript was modified as follows (line 329 and subsequent):

“Figure 17 reports the maximum temperature for both the cases. The introduction of a thermal conductive layer aims to distribute heat in the volume, mitigating the maximum temperature reached. By comparing temperature field, the use of a GNP film allows a temperature reduction of 20°C (time 1500 s) reducing the risk of TR onset.”

  1. Do the authors consider the relatively small differences in max. temp. vs. time profiles so important and relevant at some time instants so important, that the whole effort aimed at improvement of heat dissipation of the battery pack is justified? How about the costs of assemblage of relatively thin and most probably expensive material within the modules?

The presence of the GNP film changes temperature distribution within the sample, to make the comparison effective a fixed point in the middle section was chosen. Picture 17 has been updated. The introduction of a GNP film allowed to lower the temperature by 20°C.

The aim of the paper is to verify if it is possible to extract heat by a thermally conductive film, by increasing the thermal exchange surface will lead to higher improvement.

The film considered (100 microns) is made by graphite nanoplatelets and have a cost comparable to CFRP prepregs. Films are easy to integrate and flexible for reproducing the enclosure shape[1].

The availability of treatment for mitigating the maximum temperature is essential in fields such as aerospace and space where the maintenance is complex and sometimes not possible. In addition, the GNP films are a lightweight multifunctional material which could be easily integrated and could act as a barrier to fire and moisture.

[1] Fabrizia Cilento, Alfonso Martone, Maria Giovanna Pastore Carbone, Costas Galiotis, Michele Giordano, Nacre-like GNP/Epoxy composites: Reinforcement efficiency vis-à-vis graphene content, Composites Science and Technology,

Volume 211, 2021, 108873, ISSN 0266-3538, https://doi.org/10.1016/j.compscitech.2021.108873.

Reviewer 2 Report

The authors report about the development of a model coupling electrochemical – heat transfer  with the aim of predict the heat propagation in a battery pack.

The results are very interesting but the english is very bad and must be extremely improved, moreover the choice of color in the figure reporting simulation results (8, 10, 12, 15) hinders the readability. I suggest to make the color variation more homogenous all along the text.

Author Response

The authors report about the development of a model coupling electrochemical – heat transfer with the aim of predict the heat propagation in a battery pack.

We appreciate the reviewer for taking time to carefully review the manuscript and give detailed and constructive comments, which has greatly helped to improve this paper. Below is our point-by-point response to each comment.

  1. The results are very interesting but the english is very bad and must be extremely improved,

The authors have assessed the errors reported in the manuscript and have performed extensive editing of the English language.

  1. moreover the choice of color in the figure reporting simulation results (8, 10, 12, 15) hinders the readability. I suggest to make the color variation more homogenous all along the text.

Colour maps for Figures 8, 10, 12 and 15 have been modified according to reviewer suggestion. The modified pictures have been reported in this section and integrated in the new release of the manuscript.

Round 2

Reviewer 1 Report

The authors have improved their manuscript. In the present version it is ready for publication.

Reviewer 2 Report

The authors fixed the manuscript adequately

Accept as it is